# Quantitative mapping of mRNA 3' ends in *Pseudomonas aeruginosa* reveals a pervasive role for premature 3' end formation in response to azithromycin

Salini Konikkat[1], Michelle R. Scribner[2], Rory Eutsey[1], N. Luisa Hiller[1], Vaughn S. Cooper[2], Joel McManus[1]*

**1** Department of Biological Sciences, Carnegie Mellon University, Pittsburgh, Pennsylvania, United States of America, **2** Department of Microbiology and Molecular Genetics, University of Pittsburgh, Pittsburgh, Pennsylvania, United States of America

* mcmanus@andrew.cmu.edu

**Data Availability Statement:** All high-throughput sequencing data have been submitted to NCBI

## Abstract

*Pseudomonas aeruginosa* produces serious chronic infections in hospitalized patients and immunocompromised individuals, including patients with cystic fibrosis. The molecular mechanisms by which *P. aeruginosa* responds to antibiotics and other stresses to promote persistent infections may provide new avenues for therapeutic intervention. Azithromycin (AZM), an antibiotic frequently used in cystic fibrosis treatment, is thought to improve clinical outcomes through a number of mechanisms including impaired biofilm growth and quorum sensing (QS). The mechanisms underlying the transcriptional response to AZM remain unclear. Here, we interrogated the *P. aeruginosa* transcriptional response to AZM using a fast, cost-effective genome-wide approach to quantitate RNA 3' ends (3pMap). We also identified hundreds of *P. aeruginosa* genes with high incidence of premature 3' end formation indicative of riboregulation in their transcript leaders using 3pMap. AZM treatment of planktonic and biofilm cultures alters the expression of hundreds of genes, including those involved in QS, biofilm formation, and virulence. Strikingly, most genes downregulated by AZM in biofilms had increased levels of intragenic 3' ends indicating premature transcription termination, transcriptional pausing, or accumulation of stable intermediates resulting from the action of nucleases. Reciprocally, AZM reduced premature intragenic 3' end termini in many upregulated genes. Most notably, reduced termination accompanied robust induction of *obgE*, a GTPase involved in persister formation in *P. aeruginosa*. Our results support a model in which AZM-induced changes in 3' end formation alter the expression of central regulators which in turn impairs the expression of QS, biofilm formation and stress response genes, while upregulating genes associated with persistence.

under accession number PRJNA639207 and GEO accession GSE173073.

**Funding:** This work was supported by the National Institute of General Medical Sciences (grant R01GM121895) to C.J.M. and by the National Institute of Allergy and Infectious Diseases at the National Institutes of Health (grant U01AI124302 and grant T32AI049820) and by the Cystic Fibrosis Foundation Research Development Program to V. S.C. The funders had no role in study design, data collection and analysis, decision to publish, or preparation of the manuscript.

**Competing interests:** The authors have declared that no competing interests exist.

## Author summary

*Pseudomonas aeruginosa* is a common source of hospital-acquired infections and causes prolonged illness in patients with cystic fibrosis. *P. aeruginosa* infections are often treated with the macrolide antibiotic azithromycin, which changes the expression of many genes involved in infection. By examining such expression changes at nucleotide resolution, we found azithromycin treatment alters the locations of mRNA 3' ends suggesting most downregulated genes are subject to premature 3' end formation. We further identified candidate RNA regulatory elements that *P. aeruginosa* may use to control gene expression. Our work provides new insights in *P. aeruginosa* gene regulation and its response to antibiotics.

## Introduction

The Gram-negative bacterium *Pseudomonas aeruginosa* (*P. aeruginosa*) is an important opportunistic pathogen in humans. It is a leading cause of hospital-associated pneumonia [1–6]. *P. aeruginosa* infections are associated with increased morbidity and risk of death in patients with pulmonary disorders such as cystic fibrosis (CF), chronic obstructive pulmonary disease (COPD), and diffuse parabronchiolitis (DPB). *P. aeruginosa* readily forms aggregates or biofilms that are thought to exacerbate disease progression, and contribute to antibiotic resistance and the persistent nature of *P. aeruginosa* infections [7–11]. For example, in the lungs of patients with cystic fibrosis, *P. aeruginosa* biofilms cause continual damage to the airways by recurrent infection, inflammation, and airway obstruction [12]. Due to their clinical importance, understanding the regulation of biofilms has become a major focus of *P. aeruginosa* research.

Sub-inhibitory concentrations of the macrolide antibiotic azithromycin (AZM) improves clinical outcomes in patients with cystic fibrosis, diffuse panbronchiolitis (DPB) and chronic obstructive pulmonary disease (COPD) [13–17]. In mouse models of *P. aeruginosa* infection, including a CF model, AZM reduced bacterial load and improved lung pathology [18, 19]. These effects were attributed to the interference with quorum sensing (QS), suppression of virulence factors, and attenuation of host inflammatory responses. Previous studies demonstrated that AZM targets QS by an uncertain mechanism, which in turn is proposed to control biofilm formation and virulence factor production [20–29]. Consistent with these biological effects, AZM treatment changes the transcript levels of genes involved in QS, virulence factor production, and oxidative stress [2, 20–29]. For example, AZM downregulates the expression of components of the *lasI/lasR* and *rhlI/rhlR* QS systems as well as QS-controlled genes coding for virulence factors such as elastase, flagellar motility factors, and rhamnolipid synthesis [2, 20–22, 25, 27–29]. AZM also downregulates QS-regulated oxidative stress response genes including catalase *katA* and superoxide dismutase *sodB* [25, 30]. While these pathways clearly respond to AZM, the underlying mechanisms remain unclear, particularly in biofilms.

AZM binds near the peptidyl transferase center on the large subunit of bacterial ribosomes and disrupts translation elongation by hindering the passage of newly synthesized polypeptides through the polypeptide exit tunnel [31–35]. Translation inhibition is hence thought to account for QS inhibition, which in turn inhibits biofilm formation and virulence gene expression by AZM in *P. aeruginosa*. Because transcription elongation and mRNA translation are spatio-temporally coupled in bacteria [36–39], the translation stalling caused by AZM could directly alter the production of full-length transcripts from QS genes and transcriptional regulators of the stress response. The extent of reduced transcriptional processivity due to AZM

treatment remains an important unresolved question because methods used in previous studies lacked the resolution to differentiate between premature and full-length mRNA transcripts.

RNA 3' end mapping allows genome-wide identification of RNA 3'-OH ends [40] and quantitation of differential gene expression [41, 42]. RNA molecules with free 3'-OH ends include both full length and premature transcripts resulting from intrinsic or factor-dependent transcription termination, transcriptional pauses and stable intermediates produced by ribonucleases [40, 43–45]. Bacteria often employ conditional transcription termination in their transcript leaders to regulate expression in response to drugs, metabolites, and stresses [46–51]. *Cis*-regulatory elements that use premature transcription termination include transcription attenuators and riboswitches, often collectively referred to as riboregulators. Their mechanisms of action involve formation of alternate RNA structures that promote full length transcript synthesis (anti-terminator) or effect premature transcription termination (terminator). The activities of riboregulators are often mediated by interaction of the nascent transcript with small molecules, ribosomes, tRNAs, or regulatory proteins. As such, riboregulators are potential therapeutic targets, and may provide synthetic biology applications [52–55]. Since conditional premature transcription termination regulates biofilm formation, virulence, and expression of antibiotic resistance genes in other bacteria [40, 51, 56, 57], efforts to identify and characterize genes likely to be controlled by riboregulators in *P. aeruginosa* would be beneficial.

In this study, we employed a modification of the genome-wide 3' end mapping approach Term-Seq [40], which we call 3pMap (Fig 1), to characterize the transcriptional landscape of *P. aeruginosa* and the acute effects of clinically relevant sub-inhibitory concentrations of AZM on gene expression. We examined differential gene expression in planktonic and biofilm cultures of *P. aeruginosa* in response to AZM. AZM altered the expression of over two hundred genes in each condition, including key factors involved in QS, biofilm formation, virulence factor production, and stress responses. Notably, most genes downregulated in response to AZM in biofilms showed increased intragenic 3' end formation, indicating a role for the antibiotic in regulating transcription. Furthermore, AZM treatment strongly induced the peristence gene *obgE*, accompanied by a corresponding decrease in premature 3' end formation in its transcript leader. Together, our results provide novel insights into transcriptional regulation in *P. aeruginosa* and support a model in which AZM-induced changes in premature 3' end formation alter the expression of key transcriptional modulators and structural genes involved in QS, virulence, stress response, and persistence.

## Methods

### Strains and culture conditions

The effects of AZM on both planktonic cells and biofilms of a UCBPP-PA14 strain isolate (genome at NCBI GEO under accession GSE173073) were investigated. Populations from freezer stock were revived by culturing overnight in LB broth at 37°C with shaking at 300 RPM to an optical density ($OD_{600}$) of 1.0. For planktonic cells, this culture was diluted to $OD_{600}$ 0.1 in 5 mL LB broth and grown to a final $OD_{600}$ of 0.47 (~3 hours shaking at 37°C). A subinhibitory concentration of AZM (0.5 μg/ml final concentration) was added to three cultures (S1 Fig). Three cultures without AZM were used as negative controls. The cultures were incubated for an additional 20 minutes at 37°C. One volume of RNA stabilization solution (5.3M ammonium sulfate, 20 mM EDTA, 25 mM sodium citrate, pH 5.2) was added and the cells were pelleted by centrifugation at 2,500xg for 10 minutes. The supernatants were discarded and the pellets were frozen at -20°C for RNA extraction. For biofilms, a mid-log phase PA14 culture was diluted to an optical density of 0.05 into six-well plates (one plate per

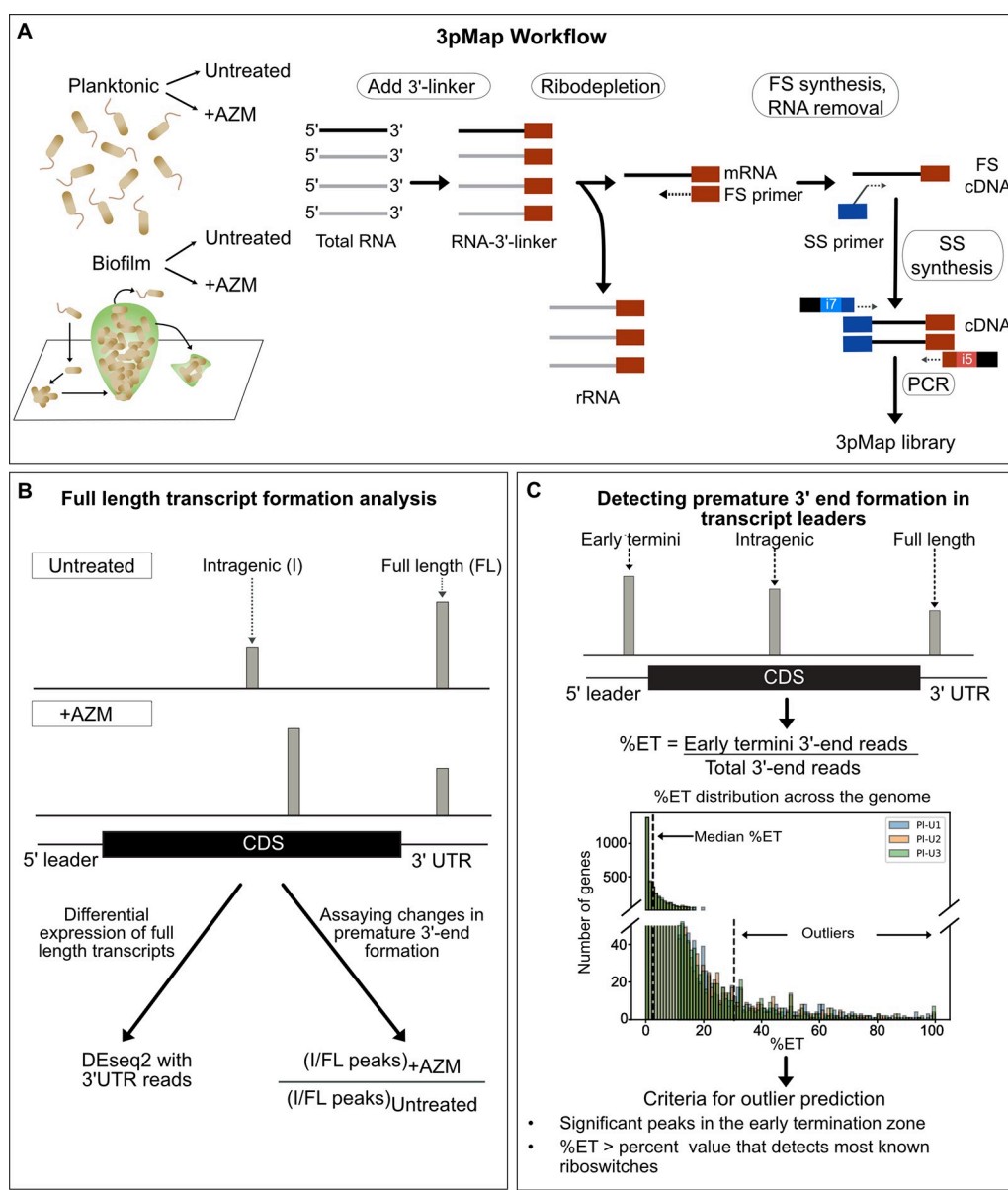

**Fig 1. 3pMap workflow and data analysis. (A)** 3pMap library synthesis. A 3' adapter is ligated to total RNA, followed by rRNA depletion. The cDNA is generated by reverse transcription with oligos containing sequences complementary to the adapter (first strand, FS) and to random priming sites (second strand, SS). The cDNA libraries are PCR amplified to generate Illumina sequencing libraries. **(B)** Assaying changes in full length transcript synthesis. 3' end reads and peaks were categorized as intragenic (I) and full length (FL) depending on their locations. Differential changes in gene expression were measured using DEseq2 of all 3' end reads (see Methods). Relative changes in premature 3' end formation were measured by comparing the I/FL peaks in AZM-treated vs. untreated samples. **(C)** Predicting RNA regulatory elements in transcript leader neighborhood. 3' end reads are assigned to genes /operons and categorized as early termini (transcript leader + 40 nucleotides) (ET), intragenic (I), and 3'UTR or full length (FL) reads. Genes with significant 3' end peaks in the ET zone and %ET >30 are considered outlier candidates.

culture) incubated at 37˚C for 24 hours. The media was removed and replaced with or without AZM (0.5 µg/ml) and incubated at 37˚C for 3 hours. The media was removed and 750 µl RNA stabilization solution was added to each well. The biofilms were scraped from the wells and the cells were pelleted by centrifugation at 2500xg for 10 minutes. The supernatants were discarded and samples were saved at -20˚C until RNA extraction.

### RNA extraction and DNase treatment

Total RNA was extracted using the Quick-RNA Fungal/Bacterial Miniprep kit (Zymo Research R2010) according to the manufacturer's instructions. To remove any contaminating genomic DNA, 20–40 μg of total RNA was incubated with 6 units of TURBO-DNase (Thermo Fisher Scientific AM2238), 1X TURBO DNase buffer, and 20 units of SUPERase-In RNase inhibitor in a 1 ml reaction for 1 hour at 37°C. The RNA was re-isolated using acid-phenol, ethanol precipitated, and resuspended in nuclease-free water. The absence of contaminating DNA was confirmed by the absence of a GAPDH PCR product, using genomic DNA as a positive control for the PCR.

## High-throughput sequencing library preparation using 3pMAP

### 3'-linker addition and rRNA depletion

To reduce secondary structure, a 5 μl reaction containing 2 μg of RNA and 375 ng of universal miRNA 3'-cloning linker (New England Biolabs S1315S) was heated at 80°C for 2 minutes and placed on ice. The RNA was ligated to the 3'-linker by incubating the mix overnight at 16°C in a 20 μl reaction containing 12.5% PEG800, 10% DMSO, 1X T4 RNA Ligase buffer, 20 units SUPERase-In RNase inhibitor, and 200 units of T4 RNA Ligase 2, truncated K227Q (New England Biolabs M0351S). The RNA was precipitated with 3 volumes of ethanol, 1/10th volume of 3M sodium acetate (pH 5.2) and 20 μg of glycogen at -80°C for 1 hour. Ribosomal RNA (rRNA) depletion was performed using the Ribo-Zero rRNA removal kit for bacteria (Illumina MRZMB126) as per the manufacturer's instructions, and ethanol precipitated as discussed above.

### High-throughput sequencing library synthesis and sequencing

Strand-specific 3pMap libraries were prepared using the QuantSeq Flex Targeted RNA-seq kit (Lexogen SKU:035.24) with the following modifications. Oligonucleotides used for library synthesis are listed below. First strand cDNA synthesis was performed as per the manufacturer's instruction using 5 μl of 2.5 μM custom first strand synthesis primer mix containing partial Illumina Read 1 adapter (below). Following the first strand synthesis, RNA was removed, and the second strand of cDNA was synthesized using SS1X mix and custom second strand synthesis primer with Illumina Read 2 sequence. Half the purified cDNA was used as a template for PCR amplification using barcoded primers provided with the kit. The library concentrations were quantified using a Qubit dsDNA high sensitivity assay (ThermoFisher Scientific Q32854) and sizes were estimated on a Tapestation high sensitivity DNA ScreenTape (Aglient). The libraries were sequenced on an Illumina NextSeq 500 platform in a single-read, 75-nucleotide run as per the manufacturer's instructions.

### Oligonucleotides for library preparation

First Strand Synthesis Mix (equimolar mix of three primers used for first strand synthesis):
  3PUTR-FS-N-Read1 5'-CACGACGCTCTTCCGATCTNATTGATGGTGCCTACAG-3'
  3PUTR-FS-NN-Read1 5'-CACGACGCTCTTCCGATCTNNATTGATGGTGCCTACAG-3'
  3PUTR-FS-NNN-Read1 5'-CACGACGCTCTTCCGATCTNNNATTGATGGTGCCTACAG-3'
  Random Second Strand Synthesis (SS) Primer
  RandomSS-Read2 5'-GTTCAGACGTGTGCTCTTCCGATCTNNNNNNNNNNNNN-3'

## Data processing

### Aligning reads to the genome and identifying significant peaks

Reads were first trimmed to remove adapter sequences. 5' ends were trimmed with a custom Perl script (5p-trimmer.pl), which removed 15–17 nucleotides preceding "CAG", then 3' ends were trimmed using Cutadapt (v. 1.9.1; parameters -a AGATCGGAAGAGC -m 25) and aligned to our UCBPP-PA14 genome sequence (NCBI GEO, see above) using Bowtie2 (v 2.1.0, default parameters) to obtain the genomic coordinates for the 3'-ends of reads [58, 59]. 3pMap is strand specific, such that reads map to the strand from which they were transcribed. We used a custom Perl script, Map3p-seq-peaks.pl, to compare aligned 3' end locations to an expected Poisson distribution of random 3' ends in 50 nucleotide windows. Candidate peaks required a raw p-value $< 5 \times 10^{-5}$, $> 5$-fold more reads than the average within the window, and a minimum of 5 total reads per peak. Strand specific peak files for each replicate were combined and RNA 3' peaks that are present in at least two of three sample replicates were identified using a combination of bedtools functions [60] (MakeSignificantPeaksFile.sh). These peaks are referred to as significant peaks in further analyses.

### Bioinformatic prediction of Rho-independent terminators in *P.* aeruginosa

Rho-independent terminators in the *P. aeruginosa* genome were predicted using ARNold, a web-tool that combines two prediction algorithms Erpin and RNAmotif [61]. Due to limitations on the input file size, *P. aeruginosa* genome was split into two halves. The genomic coordinates for predicted Rho-independent terminators in the second output file were compiled using a Python script and a mergebed bedtools operation was performed in a strand-specific manner to obtain a 6-column file with coordinates for Rho-independent terminators (S1 Table).

### Genome-browser images

Integrated Genome Viewer [62] (IGV) images were exported as Scaled Vector Graphics (.svg) images and edited on Adobe Illustrator or Affinity Designer to obtain simplified views. Views for negative strand genes were inverted using mirror display option for the ease of visualization. IGV tracks show the reads present in the same strand as the gene of interest, using autoscale. The genomic coordinates for the IGV snapshots are included in the images. 3' end read counts for samples are indicated above the tracks corresponding to the sample.

### Associating 3' end reads with *P. aeruginosa* genes

For analysing the ability of 3pMap to accurately detect RNA 3' ends, 3' end reads within 400 nucleotides of genes were associated with the coordinates in.gff file using the bedtools windowbed function. The 3' end significant peaks were associated with tRNAs and mRNAs if they were (a) intragenic located within the gene boundaries or (b) intergenic–n nucleotides downstream of the annotated 3' end and does not overlap into the next gene on the same strand (n = 400 for mRNAs and n = 100 for tRNAs as the intergenic spaces between these genes and their downstream genes were shorter). For remaining steps in the analysis, genomic coordinates of *P.aeruginosa* transcript leaders (5'-UTRs) and transcriptional unit boundaries were obtained using LiftOver (UCSC genome browser) from a previous study on a nearly identical strain [63, 64] (S2 Table). If the lifted-over transcript leaders encroached the CDS of an upstream PA14 gene, gene boundaries in the PA14 genome annotation were retained. 3' end reads within 400 nucleotides of genes were identified using the bedtools windowbed function (createFilesToAssociateWithGenes.sh). The 3' end reads and peaks were assigned to gene

transcriptional units if they satisfied either of the following conditions: a) the read is within defined operon boundaries, b) the read is downstream of the annotated 3' end of a gene and does not overlap more than 10 nucleotides into the next transcriptional unit on the same strand. The significant peaks associated with transcriptional units are identified in S3 Table (SignificantPeaksPerGeneFinder.py).

## Quantifying the effects of AZM on gene expression in *P. aeruginosa*

For differential expression, we used total read counts in the 3' UTR regions of transcriptional units (all 3' reads) other than rRNAs as discussed above. The total number of reads in the 3' UTRs was compiled using a Python script that sums up the counts for reads associated with 3'UTRs of transcriptional units (DEseqFileCreateAndRunInstruction.py), and pairwise differential expression analysis of planktonic and biofilm cultures +/- AZM were performed using DEseq2 [65] (S5 and S6 Tables). Genes with an adjusted P value (pAdj) < 0.1 were considered to have significant expression differences.

## Detecting relative changes in premature 3' end formation in the presence of AZM

Since transcription pausing or termination would result in reproducible peaks, we used reads corresponding to significant peaks of transcript ends in the intragenic (I) and 3'UTR regions (FL) of transcription units for protein coding genes for this analysis (IFLratioFileCreator_-Step1.py). To avoid false-positives due to low read counts, genes were only analyzed if the majority of the replicates have both I and FL read counts (no more than one zero value for I and / or FL per sample and treatment. Pseudocounts of 1 were used for all remaining zero values from I or FL peaks. Cochran-Mantel-Haenszel tests were used to identify genes with significant differences between the expected and observed read frequencies in triplicate samples of untreated and treated cultures (RcodeCMHIFL.R). The Benjamini-Hochberg procedure was used to control false discovery, with FDR < 0.1. Relative changes in levels of premature transcripts within the coding sequences of genes in response to AZM were estimated as the odds-ratio of intragenic (I) / full length (FL) 3' end peak read counts in AZM treated vs. untreated samples. Tests were performed in R (version 3.6.1) using default parameters in 2 x 2 x 3 matrices.

## Predicting RNA regulatory elements in *P. aeruginosa* transcript leaders

We categorized reads present in the transcript leaders or 0–40 nucleotides into the ORF as early transcript termini reads. Percentage early termini reads (%ET) was calculated as follows:

$$\% \, ET = (Early \, termini \, reads/Total \, 3' \, end \, reads \, in \, the \, gene) * 100$$

The highest % ET that detected most predicted/known riboswitches (S11 Table) in *P. aeruginosa* was employed as a cut-off (S5A Fig) [66–68]. Genes with % ET above the cut-off in at least two replicates were considered for the next step. Since significant peaks denote 3'-ends that are reproducibly present above background noise in at least two replicates, only genes with significant peaks in their early termination zone were considered as outlier candidates. The ability of the RNA sequences upstream of significant peaks to form alternate RNA structures capable of causing Rho-independent termination were analyzed using the PASIFIC (Prediction of Alternative Structures for Identification of *cis*-regulation) web-based application [69]. Sequences were analyzed for the ability to form Rho-independent (intrinsic) terminators using ARNold [61].

## Results

### 3pMap accurately identifies RNA 3'-OH ends

To investigate transcription-mediated regulatory mechanisms and the acute effects of AZM on the transcriptional landscape of *Pseudomonas aeruginosa*, we developed 3pMap. 3pMap adapts a commercially available RNA-seq kit to map mRNA 3' ends, by ligating a custom 3' end oligonucleotide adapter to total cellular RNA and directing library preparation to the adapter sequence (Fig 1A). Sequencing library synthesis using 3pMap has fewer steps compared to the Term-Seq approach [40] and requires minimal RNA input (2 μg) per sample.

We first applied 3pMap to *P. aeruginosa* PA14 grown under planktonic conditions in rich media in triplicate to evaluate its accuracy. We aligned the reads to the genome and identified 3' end peaks that are present in at least two replicates of a sample (significant peaks, see Methods). We assessed the distribution of 3' significant peaks in intragenic vs. intergenic regions of tRNAs and protein coding sequences (S2 Fig). Greater than 98% of significant peaks associated with tRNA genes were located within 3 nucleotides of the corresponding annotated 3' ends (S2A and S2B Fig). Similarly, 88.4% of significant peaks associated with mRNAs were located in intergenic regions, with <12% located within coding sequences (S2C Fig). These results show that 3pMap accurately detects known RNA 3' ends. We detected a small amount of ribosomal rRNAs retained in the samples after rRNA depletion. Alignment of significant peaks assigned to rRNAs confirmed the ability of 3pMap to detect RNA 3' ends accurately, and suggest alternative rRNA 3' ends that may be used in precursor or mature rRNA (S2D–S2H Fig). Together, these results illustrate that 3pMap identifies both known and novel RNA 3' ends.

Finally, we investigated the relative frequency of 3' ends in Rho-independent terminators. We used ARNOLD to predict Rho-independent terminators in the *P. aeruginosa* genome (S1 Table) [61]. ARNOLD predicts Rho-independent terminators bioinformatically using examples from *E. coli* and *B. subtilis*. On average, 18.4% of 3' end reads in significant peaks assigned to protein-coding genes overlapped predicted Rho-independent terminators (S4 Table). Since RNA 3' ends can be trimmed by nucleases [43], this represents a lower limit of the amount of 3' ends caused by Rho-independent termination. Thus, we estimate that roughly one fifth of 3' ends are generated by Rho-independent termination in *P. aeruginosa*, suggesting a substantial role for Rho-dependent termination.

### The effects of AZM on *P. aeruginosa* gene expression

We next investigated how AZM alters gene expression in *P. aeruginosa*, since bacterial transcription and translation are coupled [36–39]. To understand the acute response of *P. aeruginosa* to AZM, planktonic and biofilm cultures were treated with the antibiotic for 20 minutes and 3 hours, respectively. We utilized the published transcript start sites from a nearly identical strain to define transcript leaders in the strain used in this study by converting genome coordinates with liftOver (S2 Table) [63]. To assess gene expression changes in response to AZM, we performed differential expression analysis (DEseq2) using all reads from 3' UTR regions of transcriptional units (Methods; Figs 1B, 2A and 2B, and S5 and S6 Tables) with an FDR of 10% (S7 Table). In AZM-treated planktonic cultures, 209 and 68 genes showed more than two-fold up- or down-regulation, respectively (Fig 2C). In biofilm cultures, 168 genes were downregulated and 112 genes were upregulated more than two-fold (S7 Table and Fig 2C). We observe little overlap between expression changes in planktonic and biofilm cultures, potentially due to a combination of differences in the duration of AZM treatment and lifestyle-specific gene expression patterns.

In AZM treated biofilms, we observed reduced transcript levels of several global gene expression modulators associated with stress responses (*rpoS* [70, 71], *rpoH* [72], and *relA*

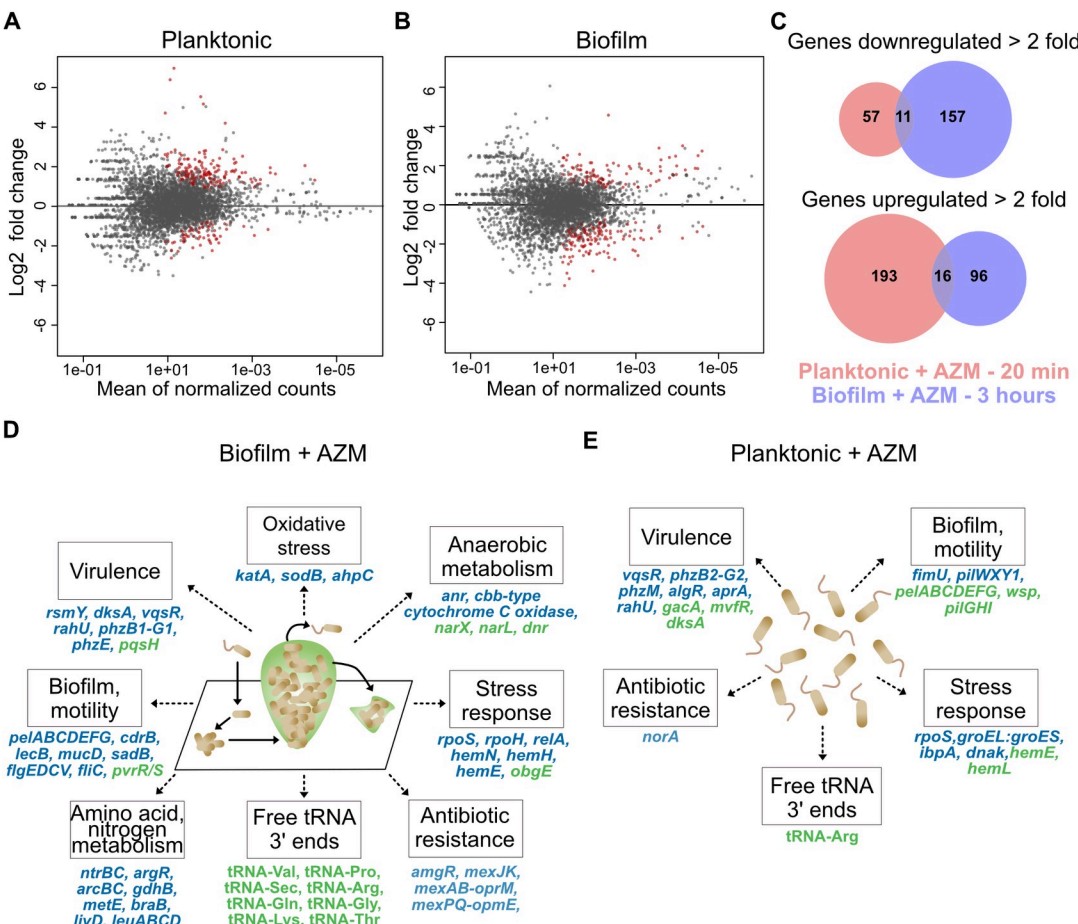

**Fig 2. AZM alters full length transcript synthesis of *P. aeruginosa* genes.** (A and B) Scatter plot of DEseq2 analysis of expression changes in PA14 planktonic cultures and biofilms treated with AZM. Red dots indicate DEseq changes at 10% FDR. (C) Genes showing >2 fold down (top) or up (bottom) differential regulation upon AZM treatment. AZM induced regulation of planktonic and biofilm genes, indicated in red and blue, respectively, show little overlap. Key PA14 genes regulated by AZM in biofilm (D) cultures and planktonic (E) cultures. Downregulated genes are colored blue, whereas upregulated genes are colored green. Many of these genes have pleiotropic functions, thus impact additional cell processes beyond the primary category in which they are shown here.

[73]), anaerobic metabolism (*anr* [74, 75]), virulence and biofilm formation (*rsmY* [76]), amino acid and nitrogen metabolism (*argR* [77]), *ntrBC* and *gdhB* [78]), QS (*vqsR* [79]), and antibiotic resistance *(amgR* [80]) (Fig 2D). The expression of stress, virulence, biofilm, motility, and metabolism-related genes, many of which are controlled by these regulators, were also downregulated. In contrast, AZM treatment in biofilms upregulated expression of the *obgE* GTPase (also known as CgtA) more than 20-fold. Previous work suggests that ObgE is a key player in the stringent response and promotes bacterial persistence [81, 82]. Overall, the functional categories in which multiple genes were differentially expressed here—inhibition of biofilm formation, AHL quorum sensing regulons, virulence, and stress responses—are broadly consistent with previously reported effects of AZM on *P. aeruginosa* [18, 24, 25, 28]. However, our results better define several genetic pathways that may underlie these changes, including those producing the stringent response (Fig 2D).

In planktonic cells, AZM also downregulated the expression of global gene expression modulators *rpoS* [70, 71], *vqsR* [79], and *algR* [83], and promoted early premature transcript

termini formation (S7 Table), but not significant downregulation, of *lasI*. As in biofilms, virulence factor genes were downregulated, though different genes were impacted (Fig 2E). These include phenazine biosynthesis genes *phzB2-G2* and *phzM* [84], *aprA* [85], *and rahU* [86]. Pilus genes *fimU* and *pilWXY1* were also downregulated, as were several heat shock proteins (*groEL*, *groES*, and *ibpA*, and *dnaK*) [87–90]. *mvfR (pqsR)*, a regulator of PQS quorum sensing [91], was upregulated, as was *obgE* though to a lesser extent than in biofilms (1.96-fold). In contrast to the biofilm samples, many genes associated with biofilm formation including *pelABCDEFG*, *gacA*, and the Wsp pathway were found to be upregulated upon AZM treatment, as well as haem biosynthesis proteins *hemL* and *hemE* [92–95]. The upregulation of biofilm genes is in contrast to previous studies and potentially reflects the relatively shorter duration of AZM treatment we employed or other differences in growth conditions.

We leveraged 3pMap data to evaluate differential changes in the levels of 3' OH ends associated with tRNAs in response to AZM. Charged tRNA carry cognate amino acids at their 3' ends, while uncharged tRNA have free 3' OH ends. Thus, 3pMap detects uncharged tRNAs by 3' adapter ligation to free 3' OH ends during sequencing library synthesis. We observed significantly increased levels of uncharged tRNAs upon AZM treatment—one tRNA (tRNA-Val) in planktonic, and twelve tRNAs in biofilm samples (tRNA-Arg, tRNA-Gln, tRNA-Gly,and tRNA-Lys, tRNA-Pro, tRNA-Sec, tRNA-Thr, and tRNA-Val) (S8 Table). Interestingly, the tRNA isoacceptors detected here pair with rare codons in *P. aeruginosa* (S8 Table). Whether the uncharged tRNAs detected arise from reduced tRNA aminoacylation, increased tRNA expression, or both in response to AZM remains to be characterized (see discussion).

## Altered premature 3'-end formation in *P. aeruginosa* genes and operons in response to AZM

Previous work reported chronic AZM treatment impairs biofilm formation by inhibiting the expression of the *lasI/lasR* and *rhlI/rhlR* QS systems [22, 25, 27]. QS inhibition is proposed to be achieved via downregulation of the *las* and *rhl* genes [22], or downregulation of enzymes responsible for QS effector synthesis [27]. The molecular mechanisms underlying AZM-mediated regulation of QS regulators have not been determined. Although *lasI*, *lasR*, *rhlI*, and *rhlR* were not significantly downregulated in our results, we observed significant premature 3'-end peaks in *lasI* and *rhlR* upon AZM treatment (Fig 3A and 3B). We used the ratio of reads at intragenic peaks to full length peaks in 3' UTRs (I/FL) as a metric for premature 3' end formation (Fig 1B) and evaluated relative differences in I/FL in AZM-treated and untreated samples using CMH tests with a 10% FDR. To avoid false-positive results, this analysis was restricted to genes with detectable intergenic and full-length 3'-ends in both treated and control samples (Methods). AZM-treatment led to altered I/FL in 550 (63.19%) and 376 (42.1%) genes in biofilm and planktonic cultures, respectively (S9 and S10 Tables). Notably, AZM treatment resulted in extensive premature 3' end formation in the 5'-end of *lasI* and a correspondingly large increase in I/FL in biofilms (~41 fold) and in planktonic cultures (~16 fold) (Figs 3A and S3A). Similarly, an ~8 fold increase in I/FL was also observed in the *rhlR* gene in AZM treated biofilms (Fig 3B). Our results show that AZM disrupts full length *lasI* and *rhlR* expression by inducing premature 3'-end formation within their coding sequences, possibly via premature termination, transcriptional pausing or accumulation of stable nuclease products.

We next investigated whether the changes in premature intragenic 3'-end formation were pervasive feature of protein-coding genes differentially regulated by AZM (Fig 1B and S9 and S10 Tables). In biofilms, 132 differentially expressed genes had detectable intragenic and full length 3' ends in both treated and control samples, and the I/FL ratios of 111 (84.0%) of these changed significantly with AZM treatment. In planktonic cultures, I/FL measurements were

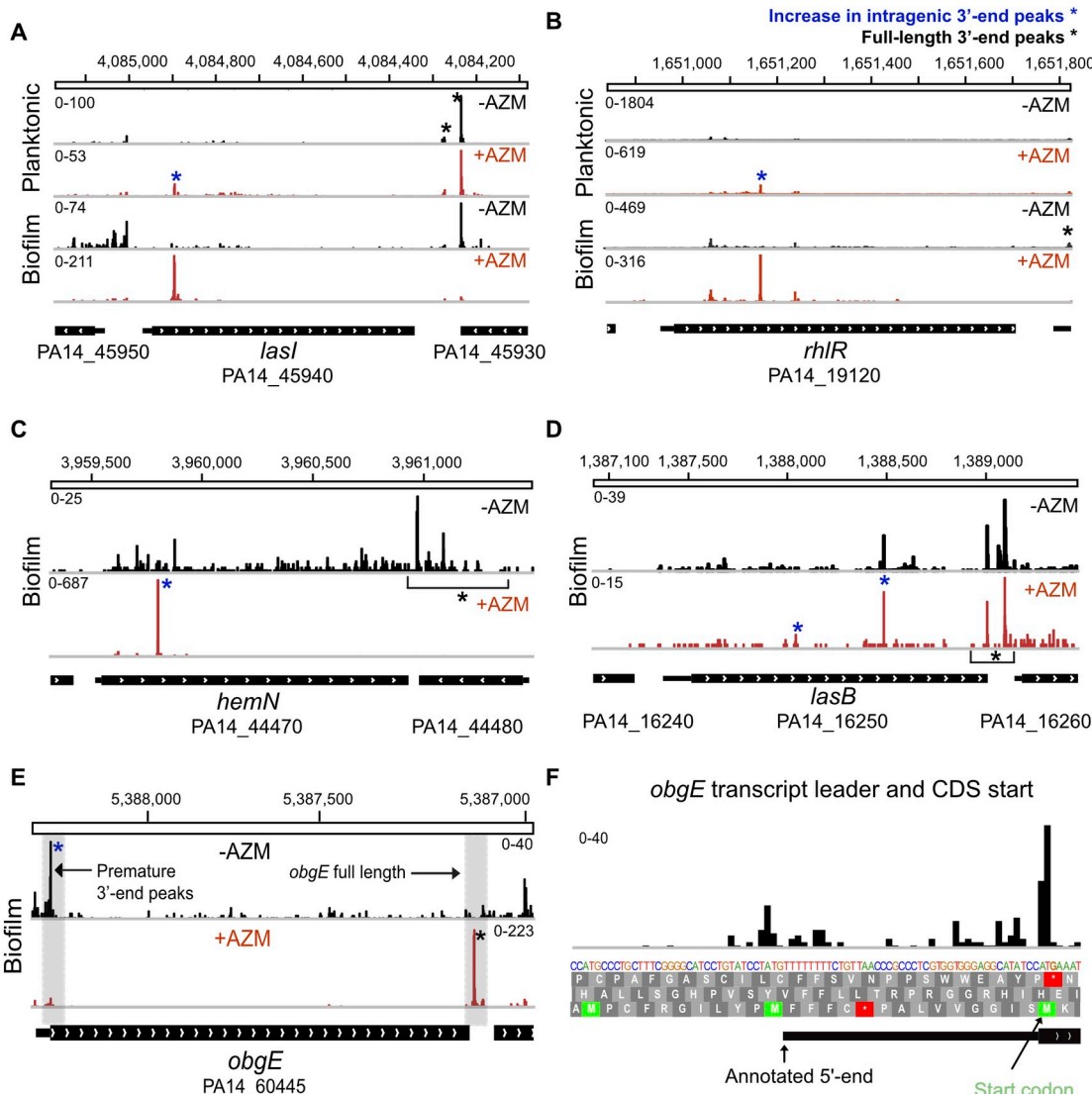

**Fig 3. AZM alters intragenic 3' end peak formation in PA14 genes. (A and B)** AZM dependent increase in intragenic premature 3' end peaks in *lasI* and *rhlR* genes. IGV views of *lasI* and *rhlR* in control (-AZM) and AZM treated (+AZM) treated samples from planktonic and biofilm cultures are shown. Intragenic peaks whose relative levels are altered by AZM are indicated with blue asterisks. The *lasI* gene in planktonic cultures and biofilm showed ~16 fold and ~41 fold increase in I/FL peak ratio upon AZM treatment, respectively. AZM treatment resulted in an ~8 fold increase in I/FL peaks for *rhlR* in biofilms. The peak is not prominent in planktonic cultures. **(C)** IGV view of the *hemN* gene downregulated by AZM (DEseq2 log₂fold change = -1.2) shows a 60 fold increase in I/FL peak ratio. The effect is less pronounced in planktonic cells **(D)** IGV view of *lasB* downregulated by AZM (DEseq2 change = ~1.95) shows moderate increase in I/FL ratio (1.74 fold) in the presence of AZM. **(E)** 23 fold upregulation of *obgE* is accompanied by a 42 fold reduction in transcription attenuation in its transcript leader. **(F)** A closer view of the transcript leader and CDS start shows termination before the rare codon for Lys at the *obgE* start codon.

available for 150 differentially expressed genes, and 76 (50.6%) changed with AZM treatment. Thus, compared to all genes, differentially expressed genes in biofilm and planktonic cultures were enriched ~1.3 fold and ~1.2 fold for changes in I/FL, respectively. Notably, the vast majority (93 of 100) of genes downregulated by AZM in biofilms had increased levels of premature 3' ends compared to full length transcripts (e.g Fig 3C and 3D), suggesting downregulated genes are 1.6-fold enriched for premature 3' end formation (Fisher's Exact Test (FET), p = 0.001913). These include global regulators *rpoS*, *anr*, and *relA*, as well as genes involved in

biofilm formation, stress response, and virulence factor genes such as *hemN* (Figs 3C and S3B), *lasB* (Fig 3D), *rahU* and *pelABCDEFG* [70, 71, 73, 74, 96]. The targets that exhibit no evidence of enhanced premature 3' end levels in their transcript leaders, yet are still significantly downregulated upon AZM treatment, may reflect expression changes in upstream regulators by AZM. The *lecB*, *argR*, and *sodB* genes are notable examples [25, 30, 77, 97].

Changes in premature 3' end formation could also reciprocally contribute to upregulation of genes in response to AZM treatment. Indeed, 18 of the 32 upregulated genes in biofilms, and 68 of the 118 genes upregulated in planktonic cultures exhibited corresponding decreases in premature intragenic 3' end formation, an 8.1-fold enrichment in biofilms and a 2.6-fold enrichment in planktonic cultures (FET p $< 7.8$ x $10^{-8}$). The most upregulated gene in AZM-treated biofilms is *obgE*, a conserved GTPase involved in the stringent response and persistence in *P. aeruginosa* [81, 82]. Notably, *obgE* also showed the highest reduction in I/FL peak ratio in response to AZM through a striking reduction in premature 3' end formation at the ObgE start codon (Figs 3E, 3F and S4). Thus, relief of premature 3' end formation occurs at most genes upregulated by AZM and may contribute to induction of the stringent response. These data suggest that premature 3' end formation contributes to the gene expression response to AZM, such that increased premature 3' end formation downregulates the expression of some genes, while decreased premature 3' end formation upregulates the expression of others.

## Predicting RNA regulatory elements that in *P. aeruginosa* transcript leaders

Transcription regulation by attenuators, riboswitches, RNA thermometers, sRNAs etc. can result in premature 3' ends in the transcript leaders and surrounding sequences [46–49, 98]. We reasoned that transcript leaders regulated by early transcription termination, pausing, and / or targeted degradation would be characterized by a high incidence of 3' end formation detectable by 3pMap. To identify genes with high incidence of premature transcript termini in their early termination zone (transcript leader + 40 nucleotides in the coding sequence) (Fig 1C and Methods), we calculated the percentage early termini (% ET) for each gene in planktonic and biofilm samples grown +/- AZM. The median %ET was $< 3.0\%$ for the tested samples (Figs 1C and S5B). As riboswitch activity has been shown to result in reproducible premature 3'-end termini, we used a heuristic cut-off of 30%, which is $>$10-fold above the median %ET and sufficient to detect most riboswitches documented on Rfam and predicted by Riboswitch Scanner to predict genes with regulatory mechanisms causing premature 3'-termini formation in their early termination zone (S5A Fig and S11 Table) [66–68]. Genes with %ET values $<$30% and lacking significant peaks representative of reproducible 3' end termini in the transcript leaders were not considered in further analysis (e.g. Fig 4A).

We identified 282 protein-coding genes with more than 30% of their 3' ends in the early termination zone (S12 Table). This included genes previously reported to be controlled by transcription termination in *P. aeruginosa* or other bacteria, e.g. *amiE* operon (Figs 4B and S5C–S5F), attesting to the ability of our approach to predict genes that are potentially regulated in this manner [99–102]. The gene with highest early transcription termination is *rhlI*, from whose transcript leader the sRNA *rhlS* was recently reported to be derived [103] (S5C Fig). We further utilized PASIFIC, a bioinformatic tool for predicting regulation via alternate structure formation and rho-independent termination to examine our new candidates for potential alternate structure formation and regulation by premature transcription termination [69], and ARNOLD [61] to separately predict rho-independent terminators (S13 Table). Forty-one premature termination candidate genes were predicted to have the propensity to form alternate

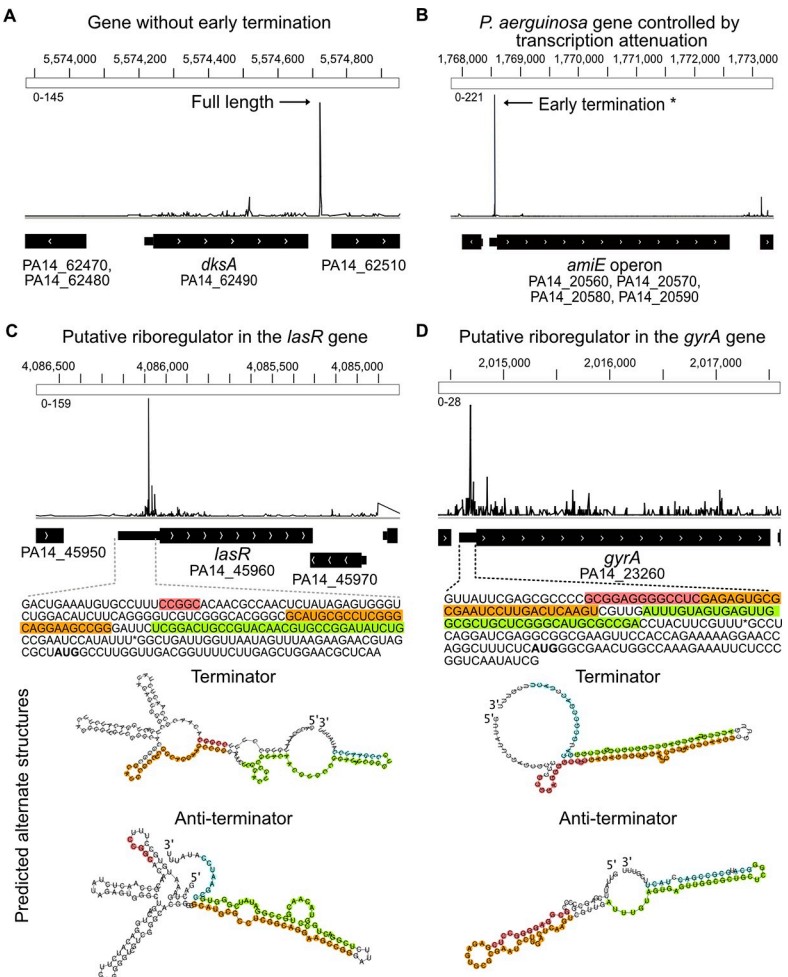

**Fig 4. Predicting genes regulated by premature 3' end formation in transcript leaders.** IGV views show strand-specific 3' end counts. (A) Example of a gene (*dksA* PA14_62490) not predicted to be regulated by premature transcription termination in its transcript leader. The major 3' end peak is in the 3' UTR. (B) IGV view of early termination in the *amiE* operon, known to be regulated by transcription attenuation, shows major 3' end peak in transcript leader. (C) IGV view of *lasR* shows premature 3' end peaks in its transcript leader. The sequence of the *lasR* early termination region and predicted alternate secondary structures in the transcript leader are shown below. The asterisk denotes a premature 3' end peak. Alternate structures with potential to effect early termination (terminator and transcriptional read-through (anti-terminator was predicted with PASIFIC [68] using the sequence upstream of the asterisk (corresponding to a 3' end significant peak) as the input. (D) IGV view of DNA gyrase A (*gyrA*) shows premature 3' ends in its transcript leader. The sequence of *gyrA* transcript leader and 40 nt in the CDS and the alternate structures predicted by PASIFIC are shown below, as in C.

structures and / or contain rho-independent terminators. Notably, these new candidates included several genes critical for *Pseudomonas* biology, such as the QS and virulence gene *lasR* [104] (Fig 4C), DNA gyrase A (*gyrA*)—associated with fluoroquinolone-resistance currently being explored as a pharmacological target (Fig 4D) [105–107], and the cytochrome c oxidase operon (not shown). Further biochemical and functional analysis of these candidate regulatory elements could provide insights into mechanisms controlling *P. aeruginosa* gene expression. In summary, we identify hundreds of genes in *P. aeruginosa* putatively regulated by early transcription termination or nuclease action, many of which were predicted to form alternate structures.

## Discussion

In this study, we used 3pMap to identify RNA 3' ends in *P. aeruginosa* and to gain insights into gene regulation in response to AZM. Our study provides new insights into the role of transcriptional regulation in the *P. aeruginosa* response to AZM. Most downregulated genes, including many key transcriptional regulators and their targets—exhibit enhanced premature 3' end formation. We also found reduced transcription attenuation in some upregulated genes upon AZM treatment, most notably inducing expression of *obgE*, a central component of the stringent response. Finally, we identified over two hundred genes exhibiting high levels of premature transcript termini in or near their 5' transcript leaders. The sequences surrounding many of the premature transcript termini are predicted to form alternate structures and/or rho-independent terminators, suggesting some may be controlled by riboregulation (S13 Table). The implications of these results for understanding *P. aeruginosa* biology and therapeutic interventions are discussed below.

Chronic *P. aeruginosa* infections are common in CF patients, and sub-inhibitory concentrations of AZM improve clinical outcomes [13–15]. AZM impairs biofilm formation due to altered expression of QS and virulence genes [21–29]. Previous work proposed the effects of AZM on QS resulted from inhibition of the synthesis of specific proteins [22, 27]. First, AZM might disrupt the translation of an unknown protein which in turn controls the transcription of autoinducer synthases *lasI and rhlI* [22]. Secondly, AZM might downregulate the expression of N-acyl homoserine lactone (AHL) synthesis enzymes upstream of *lasI* and *rhlI*, which in turn further dampen AHL production and inhibit QS [27]. However, because the assays used in these previous studies did not differentiate between full-length and truncated transcripts, neither of the resulting models consider the impact of AZM-induced translational stalling on coupled mRNA transcription. We investigated the role of transcription regulation on the *P. aeruginosa* response to AZM using 3pMap. AZM greatly elevated the levels of premature transcripts in primary QS regulators *lasI* (in planktonic and biofilm cultures) and *rhlR* (in biofilms) (Fig 3A and 3B). Thus, our results suggest AZM directly impacts expression of QS genes by regulating transcription or transcript stability.

Importantly, we found AZM induced premature transcript formation in key transcriptional regulators of biofilm formation and virulence (S7 Table). Such increases in premature transcript formation accompanied the downregulation of *rpoS*, *rpoH*, *vqsR* and *anr* [70–72, 74, 75, 79]. *Anr* regulates QS and virulence gene expression in low-oxygen environments such as biofilms [74, 75]. In QS mutants isolated from infections, *anr* plays a compensatory role in virulence gene expression. The expression of *rpoS*, an alternative sigma factor and a master regulator of *P. aeruginosa* stress response [70, 71] was downregulated by AZM in planktonic cells as well. Increased premature 3' end formation also accompanied downregulation of virulence factor genes targeted by QS (*lasB*, *phzB-G*, *pelABCD*, *rahU*) [25, 85, 86, 108]. The premature 3'-ends we discovered could result from altered premature termination, transcription pausing, or from nuclease activities. Taken together, our results support a model in which AZM-induced premature 3' end formation reduces the expression of these central transcriptional modulators (Fig 5A).

The effects of AZM on *P. aeruginosa* gene expression are reminiscent of previous work in other bacteria, which showed ribosome-acting antibiotics can alter target gene transcription by stalling translation [36, 109–112]. In *E. coli*, slowing translation with sublethal concentrations of chloramphenicol uncouples transcription from translation, thereby slowing transcription elongation rates and/or triggering premature transcription termination [37, 112]. Macrolide-mediated ribosome stalling is not uniformly distributed across genes but instead is determined by specific sequence features of nascent polypeptides [113]. In *Staphylococcus*

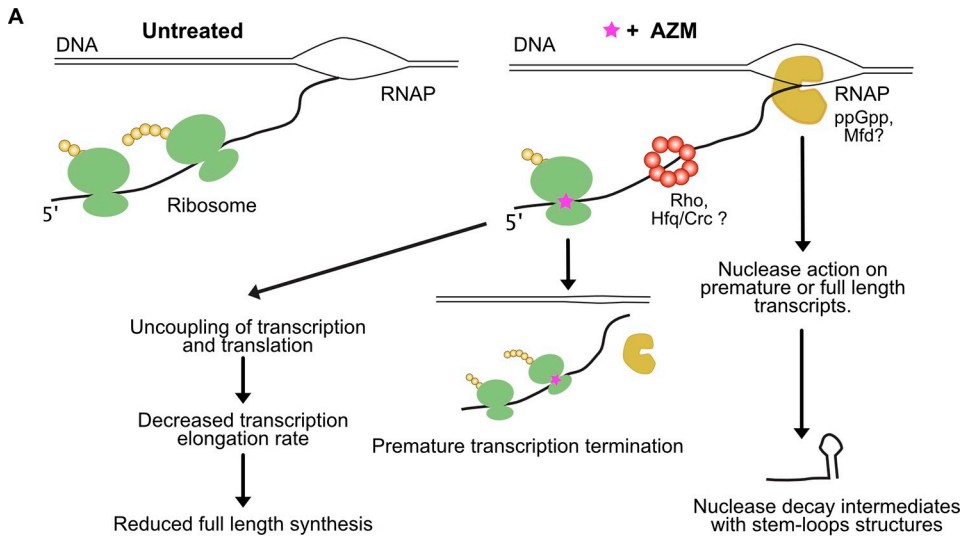

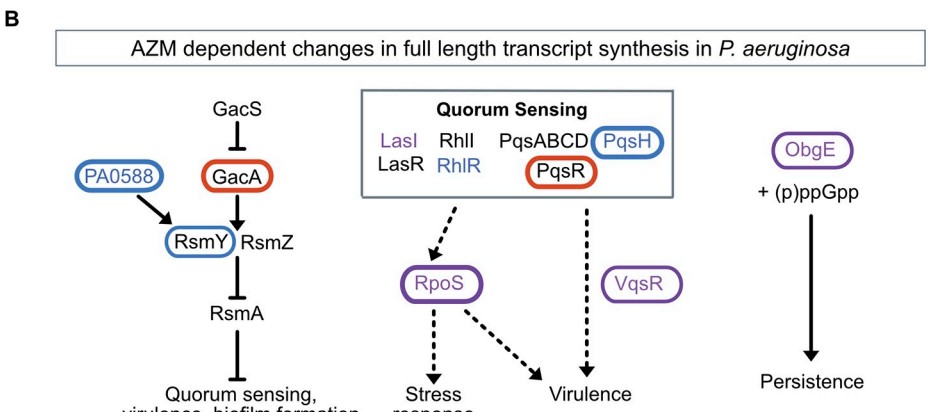

**Fig 5. Model of *P. aeruginosa* response to azithromycin. (A)** Altered premature 3'-end formation is observed in up and down-regulated genes/operons of *P. aeruginosa* in response to AZM. AZM induces changes in premature transcription termination, transcription pausing and/or transcription elongation rate, due to transcription-translation uncoupling or altered RNA polymerase pausing in response to ppGpp, thereby resulting in gene expression changes. Alternatively, nuclease action on premature and full-length transcripts could contribute to gene expression changes in response to AZM. Different molecular mediators of transcriptional control that could affect transcriptional changes are indicated–Rho and Mfd (transcription termination), ppGpp (RNAP pausing), and Hfq (sRNA action and nuclease recruitment). **(B)** Impact of AZM on gene expression of key *P. aeruginosa* regulators. Network diagram depicts several genes found to be differentially regulated and exhibit premature transcription termination within the GacS/GacA signal transduction system, quorum sensing systems, and stringent response regulators. These genes, and their regulons, have been shown to be highly interconnected, therefore only a subsection of interactions are shown here. Oval outlines indicate that a gene was differentially expressed upon AZM treatment as detected by DESeq2. Significant hits for altered relative I/FL ratio upon AZM treatment is shown by color of the text. Genes with altered expression or early transcription termination in biofilms are indicated in blue, in planktonic culture in red, and in both conditions in purple. The nature of the expression changes is indicated in Fig 3D.

*aureus*, AZM caused sequence-specific ribosome stalling in the middle or late stages of translation [32, 33]. Interestingly, AZM-specific ribosome stalling hotspots, referred to as 'macrolide arrest motifs' [114], were enriched in virulence-related and essential genes [113]. By quantifying premature transcription, we found premature transcript termini accumulate at most, if not all, downregulated genes in *P. aeruginosa*. This accumulation of premature transcripts likely reflects sequence-specific ribosome stalling. One interesting possibility is that *P. aeruginosa* has evolved macrolide-responsive stall sequences in QS and stress response genes (e.g. *rpoS*

and *lasI*) to promote survival in response to macrolides produced by environmental competitors [115]. Additional studies, including ribosome profiling, are needed to identify translation stalling motifs in *P. aeruginosa* and further investigate antibiotic induced ribosome stalling on transcription.

Bacterial transcription termination occurs via intrinsic and factor-dependent mechanisms mediated by Rho and Mfd [44, 45]. Other work has shown that sRNAs can antagonize Rho-dependent termination, e.g. at the *E. coli rpoS* gene [116]. We find that roughly one-fifth of reads in significant 3' end peaks correspond to predicted rho-independent or intrinsic terminators, suggesting most termination in *P. aeruginosa* is factor-dependent (S3 Table). AZM treatment did not considerably change the proportion of reads in peaks overlapping intrinsic terminators in biofilms and planktonic cultures, suggesting that factor-dependent termination could play an important role in *P. aeruginosa* AZM response.

It is possible that some of the altered 3'-end significant peaks observed in AZM-treated samples could be nuclease-degradation intermediates with stable structures. In bacteria, ribosome occupancy and events occurring during translation influence the susceptibility of mRNAs to ribonuclease attack [117]. Transcriptional pausing and translation-transcription uncoupling are thought to increase endonucleolytic RNase cleavage [117–119]. The cleavage products are quickly acted upon by 3' exonucleases until they encounter thermodynamically stable stem-loop structures. Consequently, the possibility of increased nuclease action on some mature and nascent transcripts in AZM-treated samples cannot be ruled out. For example, *hemN* and *rpoS* have been reported to be bound by *Hfq* in *P. aeruginosa* strain PA01 in the absence of antibiotics [120]. However, we also note that low levels of antibiotics that inhibit translation have been found to stabilize mRNAs typically associated with drug resistance in Gram-positive bacteria (e.g. *S. aureus* and *B.subtilis ermA* and *ermC* [121, 122]).

An interesting observation from our study is the differential increase in tRNA 3'-end levels in response to AZM.The increased 3'-ends for tRNAs, particularly rare codon tRNAs for Arg, Lys, Pro, Gln, Gly, Sec, and Val in biofilms (S7 Table), could arise from reduced aminoacylation or an increase in tRNA gene expression. Reduced aminoacylation could cause translational stalling at rare codons, further altering the synthesis of full-length transcripts (Fig 5A) [110, 111]. Increased tRNA gene expression, unless accompanied by proportionate increases in aminoacylation, could dilute the pool of charged tRNA. In this case, the net effect would also be increased translational stalling at these rare codons. The influence of rare codons on translation and transcription elongation rates is well documented in *E. coli* [107], and tRNA concentrations and codon decoding times are positively correlated in many prokaryotes [13, 116]. Interestingly, replacing a rare codon (AGG, frequency = 0.6 / 1000) for Arginine at the second codon position of the QS gene *rhlR* with the most frequently used codon (CGC, frequency = 48.8 / 1000) was previously shown to alleviate the inhibitory effects of AZM on virulence factor production in *P. aeruginosa* [123]. Our results suggest tRNAs targeting rare codons for Arg, Lys, Pro, Gln, Gly, Sec, and Val were more often uncharged after AZM treatment in biofilms. This decrease in tRNA charging could cause ribosome stalling at rare codons, altering the synthesis of full-length transcripts. Deacylated/uncharged tRNAs bind to the A-site on the ribosome, eventually triggering stringent response mechanism [124]. The stringent response leads to elevated levels alarmones such as ppGpp and ppppGpp, which are known to regulate RNA polymerase and help bacteria to survive under adverse conditions. Future work evaluating the role of tRNAs in the *P. aeruginosa* response to AZM would be beneficial.

AZM treatment caused upregulation of *obgE*, a conserved GTPase that binds ppGpp and mediates the stringent response and persistence under various stresses by increasing the dissociation of stalled 70S ribosomal units [82, 83, 125]. 3pMap identified a prominent 3' end

significant peak in the first codon of the *obgE* ORF (Fig 3E and 3F). AZM-induced upregulation of *obgE* coincided with significantly reduced levels of premature 3' end termini in the transcript leader, suggesting it may be regulated by a transcriptional attenuator. Ribosome stalling on upstream open reading frames (uORFs) in transcript leaders have been shown to relieve premature transcription termination of response to the macrolide lincomycin in *L. monocytogenes* [40]. Curiously, the *obgE* transcript leader contains a uORF (Figs 3F and S4), which may play a similar role in upregulating full-length transcription through attenuation. However, this attenuation-by-uORF mechanism seems insufficient because the observed premature termination occurs at the *obgE* start codon, which is followed by a rare codon for lysine, not in an intrinsic terminator. This suggests additional regulatory factors may contribute to the switch to full length transcription in the presence of azithromycin. For example, the switch to full length transcription may be independent of the uORF, instead relying on Rho, other *trans*-acting factors or uncharged tRNA. Notably, transcriptional pausing often occurs at translation start sites in *E. coli* and *B. subtilis*[126], and ppGpp has been shown to bind directly to *E. coli* RNA polymerase and alter its pausing behavior [127], which could potentially relieve premature termination in *obgE*. Future work is needed to determine the mechanisms underlying transcriptional regulation of *obgE*.

RNA-based regulatory strategies are gaining traction as antibacterial therapeutic targets [13, 52, 128, 129]. For example, the FMN riboswitch controls expression of flavin mononucleotide synthesis operons in many bacteria. Ribocil, a small molecule that targets the riboswitch by mimicking the FMN ligand, reduced bacterial growth of *E. coli* in a mouse systemic infection model [128]. Similarly, a small molecule targeting the guanine riboswitch, PC1, inhibited growth of *S. aureus* in mouse and cow infection models [129, 130]. The candidates we identified provide a first transcriptome-wide catalogue of candidate genes regulated via mechanisms targeting transcript leaders such as premature transcription termination, translation inhibition, or nuclease action in *P. aeruginosa*, which have the potential to inform experimental studies to discover 'druggable RNAs'. However, due to the compact nature of bacterial genomes, some of the candidate riboregulators may represent termination sites from upstream genes. On the other hand, many riboswitches and attenuators may still await discovery, as these elements are condition-specific and may require interactions with the host immune system or other cellular stresses for their activity. Indeed, our heuristic 30% ET threshold may miss elements that were less active in the metabolic conditions of the media used in our study. Additionally, some could produce novel sRNAs with regulatory roles, as was recently reported for *rhlI* [103]. Thus, further studies are needed to probe the transcriptional regulation of the elements identified here and to identify the full set of bona fide riboregulators in *P. aeruginosa*.

Regardless of the molecular mechanisms utilized, our results reveal that the accumulation of alternative 3' termini is a hallmark of the response to AZM by *P. aeruginosa* that may contribute to changes in QS and virulence gene expression. We propose a model in which the AZM-induced premature transcription termination, pausing, and/or nuclease action on multiple central transcriptional modulators cascades through their regulatory targets (Fig 5B). Furthermore, the AZM-dependent increase in tRNA 3'-ends and induction of *obgE* expression suggest that AZM may trigger the stringent response and persistence in *P. aeruginosa*. This implies that the efficacy of AZM as an anti-virulence treatment may trade off against the induction of persister phenotypes, a phenomenon warranting further study. In summary, we provide new insights into how AZM treatment affects the transcriptional landscape of *P. aeruginosa*, which has important implications for the development and implementation of treatments for infections.

## Supporting information

**S1 Fig. The Minimum Inhibitory Concentration (MIC) of AZM was determined by broth microdilution according to CLSI guidelines [131].** Three replicate assays were performed. Briefly, twofold microdilutions of Azithromycin dihydrate (Fisher J66740) in Mueller Hinton Broth were inoculated with $5x10^5$ CFU/mL PA14 in a 96 well plate. After 16–20 hours of incubation at 37˚C, MIC was determined to be the lowest concentration of antibiotic at which no visible growth was observed.
(TIFF)

**S2 Fig. Genome-wide mapping of RNA 3' ends in untreated planktonic cultures of PA14 using 3pMap. (A).** Location of 3pMap reads with respect to the 3' end of tRNAs (RNA end = 0 nucleotide). 98.78% of the significant peaks within or downstream of tRNAs are located within 3 nucleotides from the annotated RNA ends. **(B)** Genome browser view of distribution of 3' ends around two tRNA genes. Number of 3' end reads corresponding to each sample are denoted on the top of the sample tracks. **(C)** Location of 3pMap reads with respect to the 3' end of CDSs. RNA 3' end = 0 nucleotide. 88.4% of significant peaks associated with mRNAs were located in intergenic regions, with <12% located within coding sequences. **(D)** Location of 3pMap reads with respect to the 3' end of 23S ribosomal RNAs. 99.8% of the significant peaks within or downstream of 23SrRNA genes are located within 3 nucleotides of the annotated RNA ends. **(E)** Genome browser view of the distribution of 3' ends around representative 23S rRNA and 5S rRNA genes. **(F)** Extended 16S rRNAs in *Pseudomonas aeruginosa*. 80.54% of the detected 16S rRNAs (<10,000 reads in three replicates) had RNA 3' ends in +35 and +36 nucleotides downstream of their annotated RNA 3' ends. A recent study reported extended anti-Shine Dalgarno motifs in 16S rRNA in other bacteria, although they failed to identify such a motif in *P. aeruginosa* [132]. The major peak +35/36 downstream of 16S rRNA could represent extended anti-Shine Dalgarno motifs in *P. aeruginosa* 16S rRNA, capable of providing additional specificity or transcript preference for Shine Dalgarno motifs on mRNA, or unprocessed pre-16S rRNAs [132, 133]. **(G)** 99.9% of RNA 3' end significant peaks associated with 5S rRNAs were located -8 or -7 nucleotides upstream of their annotated RNA ends. **(H)** Alignment of 5S rRNA sequences from PA01 and PA14 genome annotations using Clustal Omega (www.ebi.ac.uk). The prokaryotic 5S rRNA sequence is well conserved and 120 nucleotides long, including in the PAO1 strain (Pseudomonas genome database). We aligned the 5S rRNA sequences from PAO1 and our PA14 using Clustal Omega (www.ebi.ac.uk), and found that the 3' end identified by 3pMap is identical to PAO1. The 3' end identified by 3pMap matches with the annotated PA01 5S rRNA 3' end.
(TIFF)

**S3 Fig. Changes in intragenic to full-length transcript ratios of *lasI* and *hemN*.** Total RNA was phenol-chloroform extracted from biofilm samples and treated with 1 unit of RQ1 RNase-Free DNase (Promega M6101) according to manufacturer's protocol to remove any residual genomic DNA. DNase-treated RNA was reverse transcribed using a 2 μM pooled mixture of reverse primers for the target genes of interest (S14 Table) and Superscript IV Reverse Transcriptase (Thermofisher Scientific 18090010). The reverse transcription reaction for intragenic and full-length transcripts were performed for 20 minutes at 55˚C. cDNA was amplified with the corresponding primer pairs (S14 Table) for targets of interest and Sybr Green PowerUp Master Mix (Thermofisher Scientific A25741) according to the manufacturer's protocol on Applied Biosystems StepOne Plus Real-Time PCR system. For each condition, three biological replicates were tested, with three technical replicates for each.
(TIFF)

**S4 Fig. Full coverage strand-specific 3pMap reads for *obgE* transcript leader.** Tracks for all three replicates of biofilm treated (AZM) and untreated (U) samples are shown. The numbers on the scale depict original, non-normalized read counts. Both AUG methionines of the potential uORF, though outside the annotated transcript leader from Wurtzel *et al.*, are covered by transcribed RNA in vivo. Note the strong accumulation of 3' ends at the *obgE* start codon, which appears as a "cliff" before coverage drops (compare to Fig 3).
(TIFF)

**S5 Fig. Early termination in PA14 transcript leaders. (A)** Determining the early termination zone. Number of known riboswitches reported on Rfam detected for each ET zone and %ET cut off considerations. The lowest ET zone consideration of transcript leader + 40 nt in CDS and the highest cut-off value of 30% detected 8/11 riboswitches reported on Rfam, and was therefore used for predicting putative riboregulators. **(B)** Box Whisker plot showing the distribution of %ET values in samples using ET zone transcript leader + 40 nt of CDS. **(C)** Early transcription termination in the *rhlI* gene detected by 3pMap [133]. Genome browser images show strand-specific tracks for the strand same as the gene of interest and in untreated planktonic samples unless specified. **(D)** Potential regulation of the *pyRG* gene by in *P. aeruginosa* by 3' end formation in its transcript leader. The *pyrG* expression is controlled by transcription attenuation in *B. subtilis* [134]. Genome browser image shows *pyrG*, *kdsA* operon. **(E)** Premature 3' end formation in *trpGDC* operon in *P. aeruginosa* biofilms. %ET values in planktonic (untreated and +AZM) and untreated biofilm samples were below the cut-off value. However, in AZM-treated biofilms (+AZM), regulation by premature 3' end formation was predicted. The gene is predicted to be regulated by transcription attenuation in *B. subtilis* [135]. **(F)** A closer view of the transcript leader and early termination peak in *trpGDC* operon. The 3' end peaks are located adjacent to a potential uORF with GTG start codon.
(TIFF)

**S1 Table. Genomic coordinates of Rho-independent terminators in *P. aeruginosa* genome predicted using ARNold webserver.**
(XLSX)

**S2 Table. Genomic coordinates of genes and operons written similar to bed format and used for bedtools and assigning 3'-end reads and peaks to transcript leaders, intragenic or operon, and 3pUTR categories.**
(XLSX)

**S3 Table. 3' end significant peaks associated with genes and operons in *P. aeruginosa*.**
(XLSX)

**S4 Table. Percentage of 3' end significant peaks associated with rho-independent terminators in the protein coding genes.**
(XLSX)

**S5 Table. Read counts associated with 3' UTRs of genes/operons replicates in untreated (U) and azithromyin-treated (AZ) biofilm(BF) samples and differential expression analysis results from DEseq2.**
(XLSX)

**S6 Table. Read counts associated with 3' UTRs of genes/operons replicates in untreated (U) and azithromyin-treated (AZ) biofilm(BF) samples and differential expression analysis results from DEseq2.**
(XLSX)

**S7 Table. DEseq2 results—Log2fold changes with pAdj < 0.1 in biofilm (BF) and planktonic (Pl) samples.** NS denotes "not significant" and represents log2 fold changes with pAdj value > 0.1.
(XLSX)

**S8 Table. tRNA genes with greater than two-fold change in 3' end reads up on AZM treatment.**
(XLSX)

**S9 Table. Relative changes in the ratio of reads in internal peaks (I) to reads in Full length peaks (FL) in planktonic samples.**
(XLSX)

**S10 Table. Relative changes in the ratio of reads in internal peaks (I) to reads in Full length peaks (FL) in biofilm samples.**
(XLSX)

**S11 Table. Rfam annotated riboregulatory elements in *P. aeruginosa* (PA01).**
(XLSX)

**S12 Table. Percent Early Termini reads (%ET) for all protein coding genes/operons in *P. aeruginosa*.**
(XLSX)

**S13 Table. Genes predicted to form alternate structures by PASIFIC or Rho-independent transcription terminators predicted by ARNold in their ET zone.**
(XLSX)

**S14 Table. Primers used for RT-qPCR.**
(XLSX)

**S1 Scripts. This file contains all of the custom scripts used for data analysis as cited in the methods section of the manuscript.**
(ZIP)

## Acknowledgments

We thank Prof. Alan Laederach and Jayashree Kumar, RNA Folding Bioinformatics Lab at University of North Carolina, Chapel Hill for advice and discussion on alternate RNA structure prediction approaches.

## Author Contributions

**Conceptualization:** Salini Konikkat, N. Luisa Hiller, Vaughn S. Cooper, Joel McManus.

**Data curation:** Salini Konikkat.

**Formal analysis:** Salini Konikkat, Joel McManus.

**Funding acquisition:** Vaughn S. Cooper, Joel McManus.

**Investigation:** Salini Konikkat, Michelle R. Scribner, Rory Eutsey.

**Methodology:** Salini Konikkat, Rory Eutsey.

**Project administration:** Vaughn S. Cooper, Joel McManus.

**Software:** Salini Konikkat, Joel McManus.

**Supervision:** N. Luisa Hiller, Vaughn S. Cooper, Joel McManus.

**Visualization:** Salini Konikkat, Michelle R. Scribner.

**Writing – original draft:** Salini Konikkat, Joel McManus.

**Writing – review & editing:** Salini Konikkat, Michelle R. Scribner, N. Luisa Hiller, Vaughn S. Cooper, Joel McManus.

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
