## [Decision Letter · Decision Letter 0]

13 Aug 2020

Dear Dr McManus,

Thank you very much for submitting your Research Article entitled 'Quantitative mapping of mRNA 3’ ends in Pseudomonas aeruginosa reveals putative riboregulators and a pervasive role for transcription termination in response to azithromycin' to PLOS Genetics. Your manuscript was fully evaluated at the editorial level and by independent peer reviewers. The reviewers appreciated the attention to an important problem, but raised some substantial concerns about the current manuscript.

The reviewers consider that the work lacks a single clear story or compelling conclusion and that the manuscript could have more impact by focusing on one specific result and demonstrating its biological relevance rather than listing different observations that popped up from their analysis. Furthermore, they noted that nothing has been followed up experimentally beyond the initial data collection, like qPCR to validate some of the most interesting results. They also suggest that mistakes and ambiguities in the data analysis should be corrected in order to make clear exactly what was done and why.

Many of issues raised by reviewers are likely addressable with clarified writing. However, some conclusions seem overstated relative to the presented data. In this case, additional experiments are required. If the authors feel that performing these experiments to shore up their speculations about mechanism are outside the scope of the study, then their conclusions should be moderated and the discussion of possible models to explain their data should be broadened.

Based on the reviews, we will not be able to accept this version of the manuscript, but we would be willing to review again a much-revised version. We cannot, of course, promise publication at that time.

If you decide to revise the manuscript for further consideration at PLOS Genetics, please aim to resubmit within the next 60 days, unless it will take extra time to address the concerns of the reviewers, in which case we would appreciate an expected resubmission date by email to plosgenetics@plos.org.

[LINK]

We are sorry that we cannot be more positive about your manuscript at this stage. Please do not hesitate to contact us if you have any concerns or questions.

Yours sincerely,

Ivan Matic

Associate Editor

PLOS Genetics

Lotte Søgaard-Andersen

Section Editor: Prokaryotic Genetics

PLOS Genetics

Reviewer's Responses to Questions

**Comments to the Authors:**

Reviewer #1: In the manuscript entitled “Quantitative mapping of mRNA 3’ ends in Pseudomonas aeruginosa reveals putative riboregulators and pervasive role for transcription termination in response to azithromycin”, the authors present collection and analysis of a 3’-end RNAseq dataset for planktonic and biofilm P. aeruginosa in the absence and presence of sub-MIC azithromycin (a translation inhibitor). The strengths of the work are a rational analysis method for identifying significant peaks from what is typically noisy data generated by 3’-end sequencing, and a number of interesting findings. These include: a DE analysis of biofilm vs. planktonic cells + and – azithromycin, analysis of how early termination may impact this response, and some observations of key genes that appear to be impacted by early termination. There are quite a few nice little stories presented here, several of which are quite interesting results.

However, the work lacks a single clear story or compelling conclusion. The titular finding is not surprising given what is known about translation inhibition in gram-negative bacteria generally. Nothing has been followed up experimentally beyond the initial data collection, not even qPCR to validate some of the more interesting results (such as the potential tRNA charging differences). The authors could have more impact by focusing on one specific result and demonstrating its biological relevance rather than listing a scattershot of things that popped up from their analysis.

My major issues with the manuscript are below, and many of these are likely addressable with clarified writing.

1. 3pMAP – authors claim this is an improved approach in the abstract. While it has fewer steps, and requires less material no data are presented to directly demonstrate that it is improved over TERMseq. In particular, since TERMseq is typically accompanied by traditional full-transcript RNAseq, one of their references (Ma et al. BMC Genomics 2019) would argue that the reverse is true. I think it probably does the job, but whether it is a second ligation step, or a PCR step with a random primer may in fact change findings, and this is not addressed by the claim.

2. The authors use the known riboswitches in P. aeruginosa to calibrate the threshold for what constitutes a significant percentage of early termination events. Of the riboswitches identified and used, how many are expected to act through transcription termination vs. other mechanisms such as translation inhibition (which could indirectly impact either rho-dependent termination or mRNA decay)? Do the authors expect these riboswitches to be in the “terminated” form under the conditions which they collected their data?

3. I am a little confused how operons and multi-cistronic genes would be impacted by the analysis method if indeed only 3’ mapping data was used for analysis. It seems that 1) all 3’ ends are captured. 2) 3’ ends are associated with a gene by mapping and can be classified as either “pre-mature” or not. What about genes which are expressed, but do not have a termination event within or immediately following them? How is expression of these genes quantified? Perhaps it doesn’t matter if AZM doesn’t impact them? The Ma et al. referenced above suggests that 3’-mapping does do better with finding shorter rather than longer transcripts, how does this square with the average transcript length expected in P. aeruginosa?

4. On the tRNA charging quantification, it seems to be it would be difficult to know just from a reduction in accessible 3’-OH, whether this is less tRNA, or inaccessible 3’-OH due to charging. Can previous data or qPCR be assessed to control for this? This actually strikes me as one of the most interesting tidbits to come out of this work, but without this control it isn’t a convincing finding.

5. The first mention of rho is on page 17 (there is a reference to rho in figure 5 as well). Discussion of whether these new termination sites are likely to be rho-dependent or rho-independent is very important. Inhibition of translation could easily cause increased rho-dependent termination in gram-negatives such as P. aeruginosa (this is likely the predominant cause of early termination observed!). Conflating rho-dependent and independent termination (as the authors do since they do not explicitly mention rho until the very end of the manuscript) is misleading.

6. Along the same lines, PASIFIC analysis assumes a rho-independent termination event (and the formation of a rho-independent transcription terminator). The rho-independent terminators shown in figure 2C are not very convincing as terminators, the expected polyU is weak in both terminator examples shown. If these are their best examples, what do the low confidence ones look like? Furthermore, PASIFIC was trained on microbiome organisms (and tested only on gram-positive organisms in the published work), these may have very different terminator profiles than P. aeruginosa. Doing this type of analysis well is difficult, but the authors might find some approaches in this review that would be helpful for identifying alternating structures that do not depend on formation of intrinsic terminators (Fukunaga and Hamada 2018 https://pubmed.ncbi.nlm.nih.gov/30689706/)

Minor issues:

1. rplJ is known to be regulated by an RNA regulator that is homologous to the one characterized in E. coli (Fu et al. 2013 https://www.ncbi.nlm.nih.gov/pmc/articles/PMC3616713/).

2. trpGCD is known to be regulated by attenuation in Pseudomonas putida, and a homologous leader is present in P. aeruginosa. (Olekhnovich and Gussin 2001 https://www.ncbi.nlm.nih.gov/pmc/articles/PMC95228/). I would double check with a deep literature dive other examples you cite as regulated in other organisms but not in P. aeruginosa, especially if E. coli regulation is known.

3. The closest relative to P. aeruginosa that contains a T-box is a delta-proteobacteria (thought to be acquired from horizontal transfer)(Kreuzer and Henkin, 2018 https://www.ncbi.nlm.nih.gov/pmc/articles/PMC6329474/). I would not invoke them in the context of P. aeruginosa repeatedly.

4. Much of the text in Figure S3 is illegible in my review copy).

Reviewer #2: In their manuscript, “Quantitative mapping of mRNA 3’ ends in Pseudomonas aeruginosa reveals putative riboregulators and a pervasive role for transcription termination in response to azithromycin”, Konikkat et al. use a modified RNA sequencing approach to describe the effects of subinhibitory azithromycin treatment on transcript 3’ ends. This is an interesting data set that has value for a broad audience. Layers of regulation affecting transcription attenuation, translation efficiency, and transcript stability have mostly been investigated in E. coli, leaving many open questions about the roles of these mechanisms in other bacteria, including important pathogens, and the experiments presented here contribute to addressing that knowledge gap. However, in places the conclusions seem substantially overstated relative to the data supporting them, and there are some questions surrounding the data analysis that could likely be resolved by providing more detail in the materials and methods and/or addressing possible errors. Major concerns are detailed below.

Data analysis:

The data analysis in a non-standard NGS experiment is non-trivial, and while the details provided in the materials and methods are appreciated, in places even more detail should be provided. Additionally, this data set has the potential to serve as a useful resource to many other researchers, since it contains genome-scale information about 3’ transcript ends. To fulfil this role to the extent possible, it would be helpful if the supplemental tables were described better in places as noted below. Finally, it would be useful to include a supplemental table describing the positions of significant 3’ end peaks for each transcript for which they could be detected – this information about putative transcription termination sites would help many researchers designing genome modifications or expression constructs to make accurate choices about transcript boundaries. Depositing the raw data in an NCBI database is appropriate and appreciated, however I might suggest that the GEO database would be the most appropriate and helpful place to deposit these data. This database allows tables of processed data to be associated with the raw data, includes standardized and detailed information about the data analysis, is manually curated, and allows private links to be provided for reviewers prior to publication of the data.

1. The authors cite Wurtzel et al. (PloS Pathogens, 2012) as the source for their transcription start site definitions, but claim that this study was performed in the PAO1 strain and describe a procedure for “transferring” the annotations to PA14. This is definitely incorrect – the Wurtzel et al. study was performed in a PA14 strain according to the published description, and it is clear from their supplemental files that the genome they used is a PA14-like genome. It is possible that there are minor differences between the genome sequence used by Wurtzel et al. and the genome sequence used in these studies. Supplementary table 7 does seem to reflect some differences in the exact nucleotide positions of transcript starts and stops relative to the Wurtzel et al. supplementary tables. This could be in part due to confusion related to strain names – the NCBI reference sequence ID listed in supplementary table 7 is for the commonly used strain whose complete name is UCBPP-PA14, and there is at least one other published isolate genome called “PA14” that is distinct. Both Wurtzel et al. and the current manuscript use only “PA14” to describe their strains, so there is some ambiguity. If the strain used here was actually UCBPP-PA14, it would be helpful to state this clearly somewhere in the manuscript. Additionally, the references to Wurtzel et al. using PAO1 should be corrected, and the reason for the described procedure to transfer annotations should be clarified. Finally, in the method for transferring the annotations, it is noted that if a 5’UTR was predicted to overlap the upstream ORF, the ORF boundary was used as the boundary of the downstream transcript. This seems unjustified. There are many examples in multiple organisms of promoters and transcription start sites falling within ORFs.

2. All of the supplementary tables seem to use transcript definitions that acknowledge some polycistronic transcripts, as is appropriate. However, the method used for defining operons is unclear. As mentioned above, supplementary table 7 seems to be derived from Wurtzel et al., but for several genes, the operon definitions used in the other supplementary tables is different from that in table 7. The reasons for this should be clarified.

3. Supplemental table 7 is described as “Bed file used for assigning 3'-end reads and peaks to transcript leaders, intragenic, and 3pUTR categories.” but does not seem to contain sufficient information for fulfilling this role – it has only a single transcript start and end position per transcription unit. The additional required information for defining these regions should be added.

4. Supplementary table 2 appears to be in a random order and has blank lines.

5. Supplementary table 6 could be better described. I assume it contains read counts for reads associated with significant peaks in the early termination zone and full length zone for each transcript and replicate, and that these values were used in the CMH statistical analysis. However, these assumptions are inconsistent with other aspects of the described analysis. The materials and methods report using a pseudocount of 1 for regions with no peak reads, but there are zeros recorded in the table. Furthermore, it seems that using a pseudocount of 1 could in some cases have a significant impact on the outcomes, because the actual counts leading to statistically significant results are quite low in some cases. Also, the counts recorded here seem to reflect wildly different percentages of early termination for some genes compared with the values reported in supplementary table 2. I understand from the materials and methods text that table 6 assessed reads associated with significant peaks while table 2 used all reads in the two regions, but both tables report assessments of the prevalence of early termination, and the reasons why they give very different numbers for some genes should be addressed. Finally, table 6 does not appear to include data for the planktonic condition, but statistical analysis of early termination in this condition is described in the text.

6. Several references to custom scripts appear in the materials and methods. Using custom scripts is often required and acceptable, but what these scripts do should be better described. For example, it is not clear if or how read counts were normalized to library depth for the DESeq2 analysis. It is also not clear if the definition of “significant” peaks strongly penalizes broader peaks with reads spread over several nucleotide positions. Such broader peaks are expected from some mechanisms for transcription termination, and could be missed if the analysis method strongly favours 3’ ends that are specific to a single nucleotide position.

Other issues

1. Results relating to rRNA transcripts cannot generally be assumed to be valid when samples have been treated to deplete rRNA. The depletion should be expected to have a significant and potentially highly biased effect on the population of rRNA molecules remaining. All 3 rRNA species are transcribed in a single transcript that is then processed, so It is not surprising to detect 3’ ends that could be related to various processing intermediates. However, the prevalence of these intermediates could be highly skewed by the depletion step, so no conclusions should be drawn. This does not seem to be a major conclusion of the manuscript and should just be removed.

2. The observation that uncharged tRNAs (free 3’-OH available for ligation to adapter) increase in the AZM-treated condition for several rare codon tRNAs is very interesting, but the relevant parameter for drawing connections to translation elongation efficiency and the stringent response is the ratio of charged to uncharged tRNAs so knowing only about the abundance of uncharged tRNA is not enough information to make these connections. It seems possible, for example, that rare codon tRNAs are simply upregulated under a condition of “translation stress” that might be caused by AZM. If this is the case, the abundance of charged tRNAs might also increase, thus increasing the overall concentration of charged rare codon tRNAs, which would have the opposite biological impact compared to the proposed interpretation. Either way, this result is interesting, but an additional experiment to assess the total abundance of those tRNAs or the ratio of charged to uncharged (by qPCR or northern blot) would significantly improve the ability to interpret the result.

3. The focus on “riboregulation”, which seems to generally refer to riboswitches and other examples of alternative RNA secondary structures playing regulatory roles, seems a bit misplaced. Computational prediction of RNA secondary structures without any experimental evidence to back those predictions up is not very reliable. Long stems in particular are suspect, given that they would require a huge length of nascent RNA to be exposed in order to form. As noted in the discussion, some of these predicted terminators could well be associated with termination of upstream genes, or even small RNAs, as was recently described in the leader of the RhlI gene. Small RNAs in general (outside a small handful of examples) are poorly annotated and little explored in Pseudomonas aeruginosa. Structures predicted from the genome sequence could also reflect functions of RNAs transcribed from the opposite strand. Concluding that any of the observed “early termination” events relate to the presence of novel riboswitches or alternative structures would require additional experiments, such as chemical probing of the RNA secondary structure or demonstration that mutations affecting the proposed structures affected the observed AZM phenotypes. The general model that AZM-induced translational stalling leads to premature transcription termination in some transcripts, and that this mechanism relies on the formation of alternative secondary structures in the nascent transcripts is not unreasonable, but the data presented do not strongly support this model, and other alternatives must also be considered. For example, many of the computationally predicted alternative secondary structures coinciding with observed 3’ ends in 5’UTR regions are far upstream of the translation start, with no indication of uORFs. This makes it difficult to understand how translational stalling would be involved. The putative uORF shown for obgE requires a transcription start site substantially upstream of the annotated one, so its relevance is highly speculative.

Other possibilities that are not addressed include: 1) cleavage of transcripts: the sequencing method used cannot distinguish between nascent transcript ends and cleavage events that leave a 3’-OH, and many bacterial ribonucleases have cleavage mechanisms that leave a 3’-OH. Many of the transcripts that were affected by AZM have also been described as targets of Hfq and/or Crc binding, and in some cases Hfq can recruit RnaseE to cleave the bound transcript. A connection between AZM treatment and Hfq/Crc mediated translational/RNA stability regulation would be very interesting, and seems at least as reasonable as a model to describe the data as the riboregulation model that is proposed. 2) intrinsic pausing behavior of the RNA polymerase: while stabilized pausing events mediated by hairpin formation in the nascent RNA are described in the literature and contribute to rho-independent transcription termination, the RNA polymerase appears to pause very frequently without hairpin formation, and many transcriptional regulators can impact its behavior in these pauses. Interestingly, such pausing events have been proposed to be especially common at start codons, due to the biophysical characteristics of the DNA duplex over the sequence context of the shine-dalgarno sequence and the AUG start codon (Larson et al., Science 2014). It is interesting that several of the observed affected premature termination events occurred at start codons. (p)ppGpp/DksA and other secondary channel binding regulators have been proposed to affect behavior at transcription pause sites, so the possibility that ppGpp could affect the outcome of a pause at the obgE start codon is an intriguing one. Obviously, additional experiments would be needed to support such a model, but it seems at least as plausible as any of the other models presented in the manuscript. 3) rho-dependent termination: in E. coli, rpoS expression has been suggested to be modulated by rho-dependent termination in the 5’UTR, which can be blocked by a small RNA. Obviously there are no data in the current study to support such a mechanism, but it is as possible as the proposed mechanisms.

A detailed mechanistic understanding of the regulatory events leading to changes in the abundances of different transcript 3’ ends may be outside the scope of this work, but the authors seem to have focused on a narrow category of mechanistic models to explain their data without any consideration of other models that are equally or even more plausible. The sentence in the discussion: “The candidate riboregulators we identified provide a first transcriptome-wide catalogue of potentially “druggable RNAs” in P. aeruginosa” in particular seems to be a dramatic overstatement of what is actually supported by the data. The manuscript could be considerably strengthened by any of a number of follow-up experiments to support or refute proposed mechanisms: for example, the 5’ end of the obgE transcript could be mapped to determine whether a uORF is in fact present; the effect of the rhlR second codon mutation described in the discussion on the premature termination behavior reported here could be measured; the impact of a relA mutant on premature termination for a gene of interest could be assessed to determine the contribution of the stringent response. If the authors feel that performing these types of experiments to shore up their speculations about mechanism are outside the scope of the study, then their conclusions should be moderated and the discussion of possible models to explain their data should be broadened.

Minor Concerns:

1. In figures 2 and 4, there are small numbers just below the left end of the genome coordinates. The meaning of these numbers should be noted in the legend. I am guessing that they describe the range of the y-axis? If so, it may be a more fair representation of comparisons between different conditions to have the y-axes scaled the same. If this obscures some of the interesting features, then the different scaling is potentially important and should possibly also be noted in the text.

2. Slowed penetration of biofilms has only been demonstrated for charged aminoglycoside antibiotics and not, to my knowledge, for azithromycin, so this is not a great justification for the increased treatment time of the biofilm cells. The authors do correctly note that the different treatment times could contribute significantly to the differences in observed effects, and it is not problematic to have two different treatments that survey a range of physiological states. However, I would recommend removing the statement about increased time required for penetration, because it perpetuates a belief that is not supported by data for azithromycin and that has been convincingly refuted for some antibiotics, including fluoroquinolones and tetracycline. Alternatively, a reference that includes biofilm penetration experiments with azithromycin should be included.

**Have all data underlying the figures and results presented in the manuscript been provided?**

Reviewer #1: Yes

Reviewer #2: Yes

PLOS authors have the option to publish the peer review history of their article (what does this mean?). If published, this will include your full peer review and any attached files.

Reviewer #1: No

Reviewer #2: No

---

## [Decision Letter · Decision Letter 1]

10 May 2021

Dear Dr McManus,

Thank you very much for submitting your Research Article entitled 'Quantitative mapping of mRNA 3’ ends in Pseudomonas aeruginosa reveals a pervasive role for premature 3' end formation in response to azithromycin' to PLOS Genetics.

The manuscript was fully evaluated at the editorial level and by independent peer reviewers. The reviewers appreciated the attention to an important topic but identified some minor issues that we ask you address in a revised manuscript

We therefore ask you to modify the manuscript according to the review recommendations. Your revisions should address the specific points made by each reviewer.

[LINK]

Yours sincerely,

Ivan Matic

Associate Editor

PLOS Genetics

Lotte Søgaard-Andersen

Section Editor: Prokaryotic Genetics

PLOS Genetics

Reviewer's Responses to Questions

**Comments to the Authors:**

Reviewer #1: In the revised manuscript entitled “Quantitative mapping of mRNA 3’ ends in Pseudomonas aeruginosa reveals a pervasive role for premature 3’ end formation in response to azithromycin” the authors present a much more focused and clear narrative and analysis of their data than exhibited by the original manuscript. The revised manuscript successfully focusses the narrative on response to azithromycin specifically. Most of my previous concerns were well addressed in the manuscript.

I only have one minor concern regarding the revised manuscript. However, as noted below, there are many problems with figure and supplementary figure mis-numbering (or just missing?) that make evaluation more challenging. I have noted some below, but the authors must go through the entire manuscript and check carefully.

My predominant minor concern is still the use of known riboswitches as a method for establishing a threshold for biologically relevant early termination. Yes, the threshold they use captures most riboswitches, but I question whether these regulators are expected to display significant early termination under the conditions examined given that most are not expected to act transcriptionally, and it is unclear what nutrient levels to expect in the rich medium conditions or what the predicted behavior of these regulators would be under such conditions (ON? OFF?, ?). Overall, this is a relatively minor concern in the context of the entire work, but I still believe this is not a robust strategy for establishing such as threshold.

Figure mis-numberings:

Line 300: “Figure 3C” I think should refer to Figure 2C?

Line 323: “Figure 3E” I think should refer to Figure 2E?

Line 409: The reference to “Supplementary Figure S3B” is incorrect.

Line 413: The reference to “Supplementary Figure S4A” is incorrect.

Line 420: Reference to “Supplementary Figure S4D-F” is not correct.

Table 1 is not included in the manuscript.

The review package only included Supplementary Tables 1-7 and Supplementary Figures 1-5. Therefore, the following calls are to missing tables/figures (and there may be additional references that I did not catch).

Line 247/248: Supplementary Figure 11A, Supplementary Table 11

Line 413:Supplementary Table 11

Line 412: Supplementary Table 12

Other areas in need of clarification:

Line 205: “…n=400 for mRNA, n=100 for tRNAs and mRNAs as the intergenic spaces….”, unclear which mRNAs use 400 vs 100. I suspect this may be a typo.

Line 269: The manuscript references three classes of genes, but I only see two (tRNA and protein coding sequences) actually mentioned. Is there a third that is missing?

Line 81: A comma after pauses in the following sentence would increase clarity: “transcription termination, transcriptional pauses and stable intermediates resulting from ribonucleases”

Line 412: typo: Predicted by Riboswitch scanner.

Reviewer #2: I would like to thank the authors for their substantial revisions to the manuscript and supplemental tables. I agree that it is greatly improved and I really enjoyed reading this new version. The dataset is of great value for many researchers and opens exciting avenues for further exploration. I note below some extremely minor typographical errors that the authors may wish to correct. Also, perhaps through a file management error, the supplemental tables that were available to me were the originally submitted ones rather than the substantially revised ones that were extensively described in the authors' response to reviewers. I believe that the authors created the new supplements and I very much look forward to exploring them. The authors and the journal should ensure that these new tables are included with the final published article.

Very minor errors:

Line 131: Probably should be 20-40 micrograms of total RNA

Line 269: “three classes of genes” but then only 2 listed. Later in the paragraph rRNA is mentioned in addition to the tRNA and protein coding sequences. This should just be made consistent.

Line 412: awkward wording/typo

Line 581: typo

**Have all data underlying the figures and results presented in the manuscript been provided?**

Reviewer #1: Yes

Reviewer #2: Yes

PLOS authors have the option to publish the peer review history of their article (what does this mean?). If published, this will include your full peer review and any attached files.

Reviewer #1: No

Reviewer #2: No

---

## [Editor Report · Decision Letter 2]

1 Jun 2021

Dear Dr McManus,

We are pleased to inform you that your manuscript entitled "Quantitative mapping of mRNA 3’ ends in Pseudomonas aeruginosa reveals a pervasive role for premature 3' end formation in response to azithromycin" has been editorially accepted for publication in PLOS Genetics. Congratulations!

Yours sincerely,

Ivan Matic

Associate Editor

PLOS Genetics

Lotte Søgaard-Andersen

Section Editor: Prokaryotic Genetics

PLOS Genetics

Comments from the reviewers (if applicable):

**Data Deposition**

http://datadryad.org/submit?journalID=pgenetics&manu=PGENETICS-D-20-01034R2

**Press Queries**

---

## [Editor Report · Acceptance letter]

7 Jul 2021

PGENETICS-D-20-01034R2 

Quantitative mapping of mRNA 3’ ends in Pseudomonas aeruginosa reveals a pervasive role for premature 3' end formation in response to azithromycin 

Dear Dr McManus, 

We are pleased to inform you that your manuscript entitled "Quantitative mapping of mRNA 3’ ends in Pseudomonas aeruginosa reveals a pervasive role for premature 3' end formation in response to azithromycin" has been formally accepted for publication in PLOS Genetics! Your manuscript is now with our production department and you will be notified of the publication date in due course.

With kind regards,

Katalin Szabo

PLOS Genetics

On behalf of:
